# Differences in Nanoplastic Formation Behavior Between High-Density Polyethylene and Low-Density Polyethylene

**DOI:** 10.3390/molecules30020382

**Published:** 2025-01-17

**Authors:** Hisayuki Nakatani, Teruyuki Yamaguchi, Mika Asano, Suguru Motokucho, Anh Thi Ngoc Dao, Hee-Jin Kim, Mitsuharu Yagi, Yusaku Kyozuka

**Affiliations:** 1Graduate School of Integrated Science and Technology, Chemistry and Materials Engineering Program, Nagasaki University, 1-14 Bunkyo-machi, Nagasaki 852-8521, Japan; bb54124102@ms.nagasaki-u.ac.jp (M.A.); motoku@nagasaki-u.ac.jp (S.M.); anh.dao@nagasaki-u.ac.jp (A.T.N.D.); 2Organization for Marine Science and Technology, Nagasaki University, 1-14 Bunkyo-machi, Nagasaki 852-8521, Japan; kyozuka@nagasaki-u.ac.jp; 3Graduate School of Engineering Chemistry and Materials Engineering Program, Nagasaki University, 1-14 Bunkyo-machi, Nagasaki 852-8521, Japan; bb52123642@ms.nagasaki-u.ac.jp; 4Graduate School of Integrated Science and Technology Fisheries Bioresources Program, Nagasaki University, 1-14 Bunkyo-machi, Nagasaki 852-8521, Japan; heejin@nagasaki-u.ac.jp (H.-J.K.); yagi-m@nagasaki-u.ac.jp (M.Y.)

**Keywords:** nanoplastics, degradation, fragmentation mechanisms, polyethylene, spherulite, cross-linking

## Abstract

High-Density Polyethylene (HDPE) and Low-Density Polyethylene (LDPE) films were used to create nanoplastic (NP) models, with the shape of delamination occurring during degradation. In the case of HDPE, selective degradation occurred not only in the amorphous part, but also in the crystalline part at the same time. Some of the lamellae that extend radially to form the spherulite structure were missing during the 30-day degradation. The length of these defects was less than 1 µm. HDPE disintegrated within units of spherulite structure by conformational defects in lamellae, and the size of the fragments obtained had a wide distribution. LDPE was synthesized by radical polymerization, so it contained a cross-linked part. The part was not sufficiently fused, and when it degraded, it delaminated and separated preferentially. The zeta potential reached a minimum value of approximately −20 mV at the degradation time of 21 days, and then increased. This complex dependence on degradation time was due to NP particle aggregation. The addition of 1% Triton(R) X-114 surfactant was effective in stabilizing the NP dispersion. The particle size remained constant at around 20 nm for degradation times of 15–30 days.

## 1. Introduction

When plastic waste enters the ocean, degradation by exposure to sunlight and mechanical stimulation by waves breaks it down into fragments less than 5 mm in size, known as microplastics (MPs), and into fine particles that pollute the environment [1,2,3,4,5]. In addition, the presence of even smaller nanoscale plastic (NP) particles in the oceans has been confirmed and is of increasing interest [6,7]. In the long history of humanity, there has never been a naturally occurring small, stable particle like NP, and it is unknown what kind of effect it has on people. Recently, research on the effects of NPs has only just begun, focusing on polystyrene latex (PSL) particles, which are easy to obtain, and/or mechanically grinded particles of several types of commodity plastic MP [8,9,10]. Some research groups have assessed the impact on ecology using NPs produced by mechanical methods [8,9]. For example, NPs prepared by mechanical grinding of polypropylene (PP) and polyvinyl chloride (PVC) dissolved in the natural surfactant (LS) on the surface of the lungs, interfering with the normal biophysical function of the LS [9]. On the other hand, PSL-NPs have demonstrated female reproductive toxicity in animal models [10]. This is due to an increase in oxidative stress and a decrease in mitochondrial membrane potential. It is also thought to have the potential to adversely affect human reproduction [10]. Furthermore, NPs are widely present in various environmental matrices, including air, water, soil, and human food, beyond the ocean. NPs can easily enter the human body by breathing, eating, or drinking, and there is a good chance that they will eventually reach the brain. NPs present in the environment can directly reach the brain. In such cases, various diseases are predicted to result from their presence in the brain. According to a review paper by Liu et al. [11], NPs can initiate a number of molecular or cellular reactions, damage the blood–brain barrier, cause oxidative stress, induce inflammatory responses, affect acetylcholinesterase activity, cause mitochondrial dysfunction, and interfere with autophagy. These articles are important in raising awareness of the dangers of NP, suggesting that NP may have a significant impact on human health. However, the NPs used are made from PSL or non-degraded plastic that has been ground up, and their shape and surface characteristics are significantly different from the actual NPs found in the environment. They are inadequate NP models for assessing the diseases that actually occur in the human body. Actual NPs are mainly composed of fragments of PP, PS, and PE (composed of HDPE and LDPE) resulting from degradation by exposure to sunlight, and are dependent on the production of industrial plastics. Since PE is the most commonly manufactured material, it is estimated that the majority of NPs in the environment come from PE products [12,13,14]. In fact, Corcoran et al. reported that PE was the predominant material in their collection of large numbers of plastic fragments from the Hawaiian coast [15]. However, the generation rate of NPs in PE materials under sunlight is very slow, so, in order to experimentally confirm the effects of NP exposure on ecosystems, it is necessary to find a way to enhance the generation of NPs in PE. In our previous study [16], we conducted PP degradation in seawater using an advanced oxidation process (AOP) with sulfate ion radical (SO_4_•^−^) as a highly efficient initiator for plastic degradation. The combination of seawater and the SO_4_•^−^ initiator resulted in the excellent acceleration of the degradation process under pH control. This accelerated degradation method can be applied not only to PP, but also to PE, which has the same autoxidation degradation mechanism.

PE can be broadly divided into HDPE, which does not have a branched chain, and LDPE, which has a long branched chain. Both PE products are commercially mass-produced, and because they are lighter than water and float on the surface of the sea, they are a major factor in the esthetic disfigurement of the sea when they are disposed of. In addition, since they are not biodegradable, they remain in the sea for a long time and are accidentally eaten by various organisms, causing serious damage to the ecosystem. On the other hand, there is also a difference between HDPE and LDPE: the presence or absence of a superstructure. HDPE, which does not have branched chains, has a high crystallinity and a spherulite structure, but LDPE, which has a low crystallinity, does not have this structure. The presence of spherulites with crystalline and amorphous regions would have a significant effect on the behavior of nanoscale fragmentation. Since PP, which is produced in large quantities after PE, also has a spherulite structure, it is expected that the behavior of nanoscale fragmentation in HDPE is similar to that of PP with a spherulite structure, and this is useful as a reference sample for preparing PP nanoparticles. In addition, the nanoscale fragmentation behavior of LDPE, which has a unique long-chain branching structure among plastics, is of interest from an academic perspective. The substantial presence of large quantities of PE-NPs raises concerns about potential risks to ecosystems, including humans. However, the quantitative impact of NP on our lives remains uncertain, and there is no definitive answer to the many concerns that now surround this issue. The development of a mass production method for PE-NPs, which are likely to be the most abundant in nature, is necessary to study the impacts of exposure to them on the ecosystem. In order to achieve this, the elucidation of the nanoscale fragmentation behavior of both HDPE and LDPE is essential for the mass production of PE-NPs.

In this study, to elucidate the nanoscale fragmentation behavior of both HDPE and LDPE, films were used to create NP models, with the shape of delamination occurring during AOP degradation with sulfate ion radicals. The shape, size, and size distribution of the fragments were measured using a scanning electron microscope and/or a dynamic light scattering (DLS) analyzer. The charge stabilities of the nanoparticles were evaluated using zeta potential measurements. The effectiveness of adding surfactants to improve NP dispersion stability was also investigated.

## 2. Results and Discussion

### 2.1. Nanoplastic Formation Behavior of HDPE Using the Accelerated Degradation Method

Polyethylene undergoes autoxidation (degradation), which causes polymer chains to break and oxidize, resulting in the formation of carbonyl compounds such as ketones [17]. Figure 1 shows comparisons of the degradation time dependences of the carbonyl index (CI) before and after the detachment of HDPE samples. The number of replicates used for the analysis was three samples each. The processing was also carried out simultaneously, and the measurements were also taken almost simultaneously (within about 30 min). With regard to statistical significance, this is a commonly used method for measuring the CI value of the polymer. The CI values of both films before and after the detachment process increased from about 0.3 to 0.7 as the degradation time increased. The maximum difference in the CI values between the two films was about 0.2 at the degradation time of 12 days. The difference was small, about 0.1, for the other degradation times, and the overall CI was higher after the detachment compared to before it, except at 30 days. This behavior suggests that the degree of oxidation of the detached part is lower than that of the remaining matrix.

The SEM photographs of 12-day degraded HDPE films before and after detachment are shown in Appendix A. The SEM images prior to the detachment show etching due to degradation, resulting in a distinct lamellar texture. In the SEM image after the detachment, the lamellar texture is not observed, and, instead, remnants of the spherulite structure can be seen. The initiation and progression of the degradation reaction require radical species to serve as initiators and oxygen molecules to allow for the degradation to proceed. Therefore, compared to the crystalline phase, the degradation selectively occurs in the amorphous phase, which is easier to penetrate and diffuse. The distinct lamellar texture observed in Appendix A is due to the selective degradation of the amorphous part. It appears that selective degradation occurs not only in the amorphous part, but also in some of the crystalline parts at the same time. The remnants of the spherulite structure, which represent the higher-order structure seen in Appendix A, show that the lamellar crystal part of the skeletal structure has been partially degraded by the degradation, making it brittle. In the 30-day film, which has been degraded longer, some of the lamellae that extend radially to form the spherulite structure are observed to be missing, as shown in Figure 2A. The length of these defects is often less than 1 µm, and the detached pieces can be considered NP. Defects in larger spherulite structural units are observed after the detachment process. This specific exfoliation behavior suggests that some lamellar parts that form the spherulite structure are susceptible to degradation and selectively break and detach. As shown in Figure 2A, the lamellar crystals of HDPE grow while twisting in a helical shape [18]. Semi-crystalline-like HDPE is composed of crystalline and amorphous phases. Loose intracrystalline chains exist within the lamellar crystals of HDPE [19]. The areas where these loose chains exist would be defective areas where radicals and oxygen can enter more easily than other regions formed by tight chains. Due to the shear stress involved during the detachment process, the spherulites with many defects in these lamellae break off in parts of the spherulite structure, causing detachment, as observed in Figure 2B. In semi-crystalline polymers with a spherulite structure like HDPE, as shown in Figure 3, conformational defects in the crystalline phase cause loose chains in the crystal, which become the starting point for fracture during the degradation, resulting in partial cracking and detachment of the lamellar crystal. Appendix A summarizes the melting point, fusion enthalpy, and crystallinity of each sample at each degradation time. These values of both samples before and after detachment gradually decrease with increasing degradation time. The behavior after the decrease in melting point of both is roughly the same, but the point where the melting point of the detached sample after 30 days of degradation is 6 °C lower than that of the undegraded sample is particularly striking. The change in fusion enthalpy, that is, the change in crystallinity, shows an oscillation range of less than 10%, except for the sample with 24 days of degradation before detachment. As the degradation time increases, the slight decrease in melting point and crystallinity suggests that the thickness of the crystalline lamellae is decreasing and that the amorphous phase is being selectively removed. The results of the melting point and crystallinity change behavior support the inference that the clear lamellar texture seen in Appendix A is due to the etching of the amorphous phase. In addition, the conformational defects in the crystalline phase (lamellae) lead to the collapse of the lamellae. Such complex melting point and crystallinity reduction behaviors result from both selective etching and conformational defects. In addition, the reason for the significant decrease in melting enthalpy of the degraded sample on the 24th is unclear, but it is likely that there are parts that have degraded so much that even the crystalline parts that have a high melting point have been destroyed. If the degradation time is even longer, it is thought that these parts will no longer have a melting point and will not be observed. Since HDPE disintegrates in units of spherulite structures by the conformational defects in lamellae, the size of the fragments obtained has a wide distribution, as shown in Figure 4. There is no clear degradation time dependence on the fragmentation size, and the errors are large. HDPE is a semi-crystalline polymer with a higher-order structure called a spherulite structure, so, in the process of degradation and fragmentation, fine fragments are generated from the amorphous, crystalline, and spherulite regions that make up the polymer. This difference in the generation matrix results in the generation of fine particles with a wide range of sizes from nano- to micro-sized, independent of the degradation time.

### 2.2. Nanoplastic Formation Behavior of LDPE Using the Accelerated Degradation Method

Figure 5 shows comparisons of the degradation time dependences of the carbonyl index (CI) before and after the detachment of LDPE samples. Compared to the CI values of HDPE, those of LDPE are about 15–85% higher for the same degradation time. PE is synthesized by two different polymerization methods [20]. LDPE is synthesized by radical polymerization using oxygen radicals as initiators. HDPE, on the other hand, is synthesized by catalytic coordination and anionic polymerization. The higher CI value of LDPE than HDPE is due to the addition of oxygen radicals to the polymer chain and indicates that the degraded LDPE has a higher degree of oxidation. In addition, as shown in Appendix A, the melting point and crystallinity are significantly lower than those of HDPE. The difference in thermal properties is due to the long chain branching of LDPE. It is known that nanosized fine particles tend to aggregate due to the Van der Waals attractive forces [21]. In order to overcome such attraction, it is necessary to have a repulsive force (repulsion) due to ionic charge. To increase the ionic charge of a polymer, it is necessary to increase its degree of oxidation (i.e., degree of degradation). It appears that it is easier to obtain NPs with superior nano-dispersion stability from degraded LDPE with a higher degree of oxidation than from degraded HDPE. As shown in Figure 6a, one large pit with a length of 4 μm and several relatively small pits with a diameter of about 500 nm can be observed. The missing parts corresponding to these pits were also captured [see Figure 6b]. The shapes of the captured parts are not granular [15,22], as typically seen in samples that have peeled off due to degradation, but rather have an irregular shape, as often seen in samples of cross-linked rubber that have partially melted. It is known that there are cross-linked parts in LDPE because a cross-linking reaction occurs during radical polymerization [23,24]. The cross-linked parts do not melt completely when heated, but remain unmelted in a gel-like form as fusion defects like ultra-high molecular weight PE [25]. Therefore, the fully molten part and the gel-like part do not melt together during melting, and a separated structure is formed, and an interface is formed after cooling. Since the interface is often inferior in terms of resistance to degradation, when it degrades, the interface degrades selectively, and the gel-like part delaminates as if it were punctured. The gel-like part is formed by the addition of oxygen radicals that occur multiple times within a single polymer chain, so it is expected to have a high degree of oxidation. This higher degree of oxidation leads to an increase in ionic strength, which will improve the dispersibility of LDPE nanoparticles. Figure 7a shows a comparison of the degradation time dependence of the zeta potential. The zeta potential reaches its minimum value after 21 days of degradation. The error also shows the largest fluctuation at each degradation time. After 30 days of degradation, the zeta potential value was approximately −10 mv, which was 10 mv higher than the −20 mv value after 21 days. NP aggregation is responsible for such complex behavior of zeta potential values over the degradation time. As the concentration of NP increases, their Debye length decreases, making them more prone to aggregation. As the degradation time increases, the amount of NP produced increases, so it seems that the concentration increases and NP aggregation progresses. There is a limit to the amount of oxidation that can be increased as a result of degradation, and it seems difficult to prevent aggregation due to increased concentration. Li et al. reported that the addition of a surfactant is more effective in stabilizing nano-dispersions than increasing the degree of oxidation (ionic strength) [26]. The result of the degradation time dependence of the particle size to which a surfactant [Triton(R) X-114: concentration 1%] is added is shown in Figure 7b. Compared to the sample without surfactant, the sample with it showed a significant difference in particle size at long degradation times of 15, 21, and 30 days. When the surfactant is not used for the longest decomposition time of 30 days, the average particle diameter is 2.5 times greater than when surfactants are used. In addition, the error in particle size is even greater than that for the 15- and 21-day degradation times. On the other hand, in the samples used the surfactant, the particle size was constant at around 20 nm for degradation times of 15–30 days, and the error was considerably smaller. Clearly, the dispersibility of nanoparticles is greatly improved by the presence of surfactants. Kokalj et al. reported that particles as small as 30 nm are present in undegraded LDPE [27]. This particle size is close to the 20 nm gel-derived particles we obtained. LDPE contains the gelatinized part of uniform size. This part is selectively separated during degradation, resulting in the production of uniform NPs of approximately 20 nm in diameter. These NPs have a relatively high degree of degradation, but they exhibit aggregation behavior in the absence of a surfactant. The nano-dispersion stability of nanoparticles derived from both HDPE and LDPE is low, and they are expected to aggregate in the absence of surfactants. These results suggest that the presence of a substance that acts as an effective surfactant is essential for the stable existence of nanoparticles in nature.

## 3. Materials and Methods

### 3.1. Materials

HDPE and LDPE were obtained from Sigma-Aldrich Co. LLC (St. Louis, MI, USA). Their melt index (190 °C/2.16 kg) values were 10 g/10 min (HDPE) and 25 g/10 min (LDPE). Potassium persulfate (K_2_S_2_O_8_) and Triton(R) X-114 were obtained from Wako Pure Chemical Industries (Osaka, Japan). The seawater used was artificial (Gex artificial saltwater). It was purchased form Amazon.co.jp (Tokyo, Japan. https://www.amazon.co.jp/gp/product/B09R9x4Y43/ref=ppx_yo_dt_b_asin_title_o02_s00?ie=UTF8&th=1, accessed on 25 August 2024).

### 3.2. Degradation Using Sulfate Ion Radicals in Seawater (Accelerated Degradation Method)

The test specimens for HDPE and LDPE films were 30 mm × 30 mm × 0.060 mm. The films were obtained by compression molding at 180 °C under 10 MPa for 11 min, and were degraded in seawater using a sulfate ion radical in an accelerated degradation method [16,28]. The procedure was in accordance with our previous reports [16,28]. (1) Several pieces of each film were put into a 100 mL glass vessel containing 20 mL of seawater solution with 0.54 g K_2_S_2_O_8_ at ca 65 °C for 12 h under stirring with a stirrer tip speed of ca 100 rpm. (2) An equal amount of K_2_S_2_O_8_ seawater solution was added to compensate for the consumption of the oxidant, and its degradation was carried out for 12 h under the same conditions. (3) The five pieces of the film were then transferred to a new 100 mL glass vessel containing 20 mL of seawater solution with 0.54 g K_2_S_2_O_8_, and the degradations were started again under the same conditions. The enhanced degradation method was run for a given number of days using (1) to (3) as one set. There were 28 sets in total. The pH value of the solution was changed from 8.2 to 3 during each set (the pH of the seawater was initially 8.2, and the SO_4_•^−^ was gradually converted to SO_4_^2−^, reducing the pH of the seawater solution to 3 at the time of the daily exchange [16]). The degraded films were used as “degraded HDPE and LDPE”.

### 3.3. Fragmentation Method

After the seawater degradation was performed using the accelerated degradation method, the fragmentation samples were obtained by detaching the degraded surfaces using vibration from a vortex mixer (vortexer) purchased from Heathrow Scientific^®^ LLC (Vernon Hills, IL, USA). The two sheets of degraded film (1 × 1 cm) were put into a conical tube (φ30 × 115 mm) with 5 mL pure water, and then the conical tube was then shaken using the vortex mixer at a speed of 3000 rpm for 1 h.

### 3.4. Characterization and Analysis

The transform infrared spectra of 16 scans were measured with a Fourier transform infrared spectrometer Jasco FT-IR 660 plus (Jasco, Tokyo, Japan) with a resolution of 4 cm^−1^ over the entire mid-IR range (400–4000 cm^−1^). The carbonyl index (CI) was calculated as the band intensity ratio of the carbonyl group (ca. 1714 cm^−1^)/scissoring CH_2_ group (ca. 1463 cm^−1^) [29].

Scanning electron microscopy (SEM) analysis was performed with a JSM-7500FAM (JEOL, Tokyo, Japan) at 5.0 kV. The working distance was approximately 3 × 4 mm. Samples were placed in a drying oven maintained at 27 °C for 30 min and sputter-coated with gold before SEM imaging.

Differential scanning calorimetry (DSC) measurements were performed using a SHI-MADZU DSC-60 Plus (SHIMAZU, Kyoto, Japan). Samples of approximately 5 mg were sealed in aluminum pans. The measurement of the samples was carried out at a heating rate of 10 °C/min under a nitrogen atmosphere. The data were taken on the 1st run. The crystallinity of PE (HDPE and LDPE) was calculated by dividing the measured fusion enthalpy (Δ*H*) by the equilibrium enthalpy (Δ*H*_0_ = 290.7 J/g) [30].

The hydrodynamic size and zeta potential values were determined under pure water suspension at 20 °C using an ELSZ-2000ZS dynamic light scattering (DLS) analyzer (Otsuka Electronics, Osaka, Japan). To determine the hydrodynamic size of the fragment contained in the conical tube after processing, approximately 2.5 mL of the suspension was placed in a borosilicate glass standard fluorescence cell (path length 10 mm, path width 10 mm, capacity 3.5 mL), and the DLS analyzer was used to measure the range 0.6 nm–10 µm using pure water (viscosity 0.8878 cP, refractive index 1.3328) as the solvent at a measurement temperature of 20 °C over 25 accumulations for each sample.

## 4. Conclusions

The naturally occurring form of NP was not spherical like PSL, but had a complex shape such as plates or fibers, and was the fragment produced by degradation. The shape was that of a delaminated section, and was used as a target for the size and shape of the model NP. The distinct lamellar texture due to etching was observed in the SEM images of 12-day degraded HDPE films. On the other hand, the lamellar texture was not observed after detachment; instead, remnants of the spherulite structure were seen. The selective degradation occurred not only in the amorphous part, but also in part of the crystalline regions at the same time. The remnants of the spherulite structure showed that the lamellar crystal part of the skeletal structure had been partially degraded. In the 30-day degraded HDPE, it was observed that some of the lamellae that extend radially to form the spherulite structure were missing. The length of these defects was less than 1 µm. The specific exfoliation indicated that some lamellar parts were susceptible to degradation and selectively broke and detached. Loose intracrystalline chains within the lamellar crystals were more susceptible to degradation and were, therefore, selectively cleaved. The spherulites with many defects in these lamellae broke off in parts of the spherulite structure, causing detachment. HDPE disintegrated into units of spherulite structure due to the conformational defects in lamellae, and the size of the fragments obtained had a wide distribution. On the other hand, the behavior of LDPE in terms of its fragmentation to nano size was significantly different from that of HDPE. The CI of LDPE was about twice that of HDPE, and it was more susceptible to degradation over the same period. Since a higher degree of degradation resulted in the higher ionic strength, it was expected that LDPE would produce more stable NP particles in terms of dispersion stability. In addition, LDPE was synthesized by radical polymerization, so it contained a cross-linked part. This part was not sufficiently fused, and when it degraded, it delaminated and separated preferentially. The zeta potential value showed a minimum value of approximately −20 mV at the degradation time of 21 days, and then increased. This complex dependence on degradation time was due to NP particle aggregation. There was a limit to how much oxidation could be increased due to degradation. The addition of Triton(R) X-114 surfactant was effective in stabilizing the NP dispersion. The particle size was constant, at around 20 nm for degradation times of 15–30 days, and the error was considerably smaller. The nano-dispersion stability of natural LDPE was low, and particles aggregated in the absence of surfactants. In order to develop a mass production method for PE-NPs, which are believed to be the most abundant in nature, it was essential to elucidate the fragmentation behavior of HDPE and LDPE at the nanoscale. The results of this study will contribute to the development of a mass production method for PE-NPs.

## Figures and Tables

**Figure 1 molecules-30-00382-f001:**
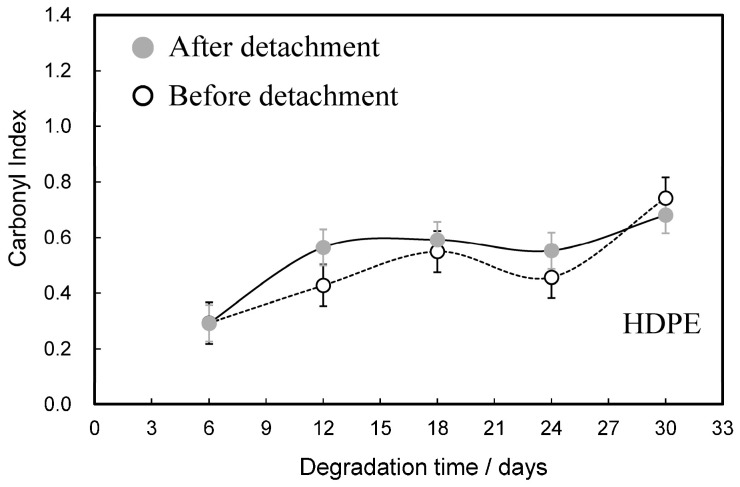
Comparisons of degradation time dependences of carbonyl index before and after detachment of HDPE samples.

**Figure 2 molecules-30-00382-f002:**
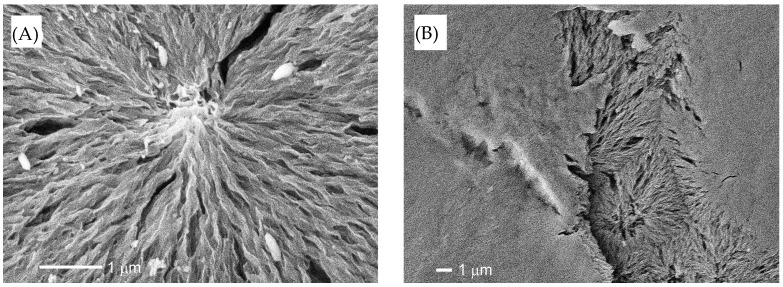
SEM photographs of HDPE samples degraded for 30 days: (**A**) Before detachment (×20,000). (**B**) After detachment (×5000).

**Figure 3 molecules-30-00382-f003:**
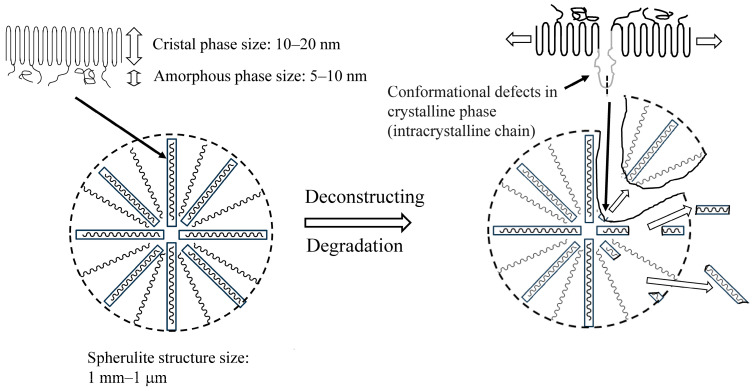
Predicted deconstruction diagram of spherulite structure.

**Figure 4 molecules-30-00382-f004:**
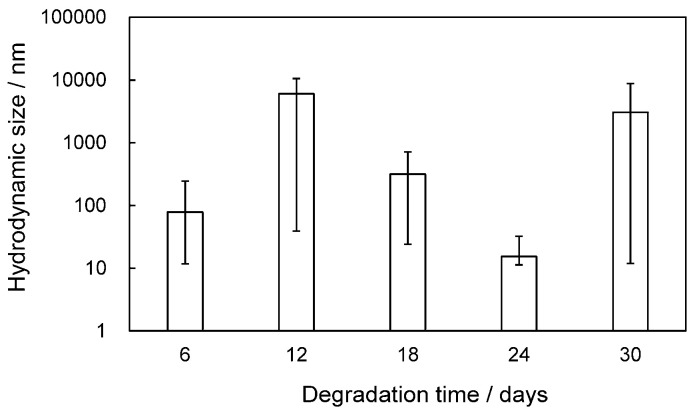
Degradation time dependence of hydrodynamic size of HDPE fragments collected after delamination treatment. Error bars indicate the difference from the mean.

**Figure 5 molecules-30-00382-f005:**
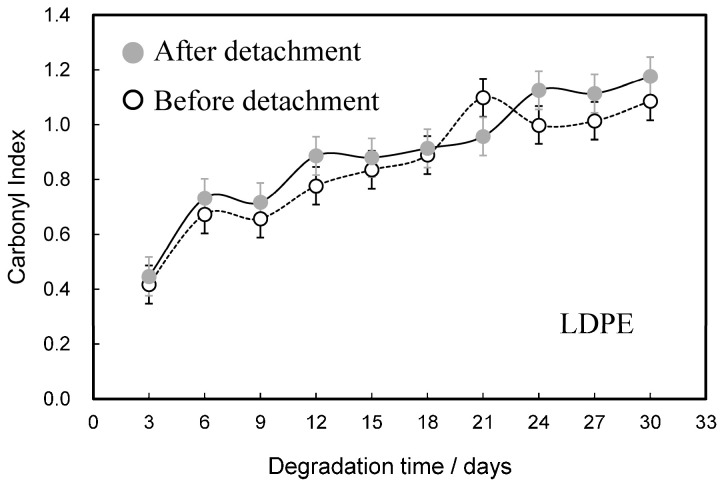
Comparisons of degradation time dependences of carbonyl index before and after detachment of LDPE samples. Error bars indicate the difference from the mean.

**Figure 6 molecules-30-00382-f006:**
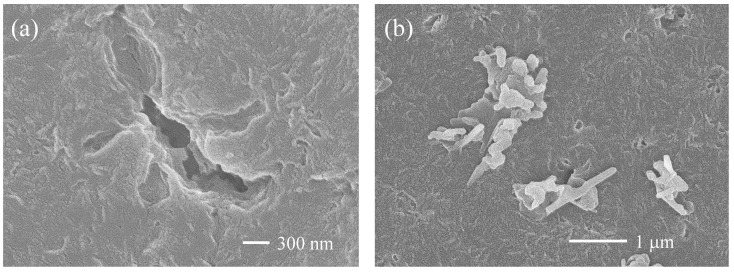
SEM photographs of detachment LDPE samples after degradation for 24 days: (**a**) surface pits (×30,000), and (**b**) recovered fragmentations (×20,000).

**Figure 7 molecules-30-00382-f007:**
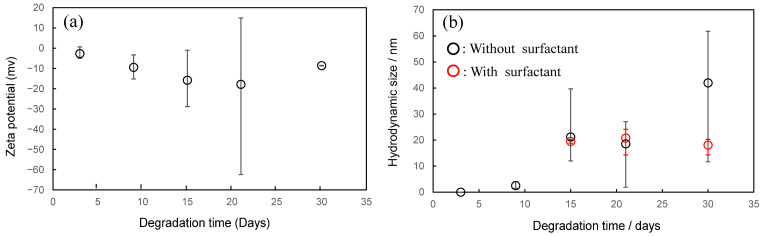
Comparison of (**a**) degradation time dependence of zeta potential, and (**b**) hydrodynamic size values after detachment of LDPE samples after degradation.

## Data Availability

Data are contained within the article.

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
