# Peer review of "Differences in Nanoplastic Formation Behavior Between High-Density Polyethylene and Low-Density Polyethylene"

_molecules, 2025, doi:10.3390/molecules30020382_

Round 1
Reviewer 1 Report
Comments and Suggestions for Authors
This manuscript deals with the degradation of HDPE and LDPE films and the subsequent formation of their nanoplastic particles. The authors compare the behavior of HDPE and LDPE and the resulting NP particles using different techniques. The introduction is clear. Results and discussions can be improved. I can only recommend the publication of this work after further major changes.
A few comments and questions are added:
1) It is usual that the Results and Discussion section is after the Materials and Methods section. I believe this is just a typographical error.
2) The authors commented on the change in fusion enthalpy for the HDPE sample of less than 10%, except for the sample with 24 days of degradation before detachment. Do they have any explanation for this exception (value 120)? Why can we observe a decrease in crystallinity in this sample?
3) The authors reported 2 different methods of synthesis (polymerization) of PE. Can the authors comment on whether there is any possible effect of the synthetic route used on the degradation process, other than the different final PE structures?
4) For a better comparison, in addition to the two SEM photographs already shown in Figure 6, I would prefer to see an SEM photograph of the surface pits of the scale of 1 μm.
5) In line 208, the sentence “After 30 days of degradation, the zeta potential value was approximately −10 mv, which was 10 mv higher than the −20 mv value after 21 days” is repeated.
6)
7) The authors only showed the results of the degradation time dependence of Zeta potential and hydrodynamic size values (with or without surfactant) of the detachment LDPE sample after degradation. It would be nice to compare these dependencies with HDPE.
8) The authors mentioned that the pH values of the solution were changed from 8.2 to 3 during each set, can they better explain the reason and possibly use a citation for this procedure?
9) The authors should explain the EDM abbreviation.
10) The authors should add the FTIR spectra to the SI section.
11) In line 275 it is written PP, there should probably be PE.
12) If the authors want to use HDPE and LDPE films to create NP models, what do they think about the fillers that are usually used in PE materials? Can they affect PE degradation or NP formation?
13) The authors wrote about the addition of 1% Triton® X-114 to stabilize the NP particles. Can the authors comment on why they use a 1% solution? Do they have a reason for choosing this concentration? Did they choose this concentration based on some publication (please add the citation) or did they try multiple concentrations to see which concentration is already suitable for stabilization? Did the authors also try different surfactants? Did the authors filter the surfactant solution before the DLS measurement? Isn't the 1% solution too viscous?
Author Response
Reviewer 1
This manuscript deals with the degradation of HDPE and LDPE films and the subsequent formation of their nanoplastic particles. The authors compare the behavior of HDPE and LDPE and the resulting NP particles using different techniques. The introduction is clear. Results and discussions can be improved. I can only recommend the publication of this work after further major changes.
Answer:We appreciate the reviewer's appropriate comments. We have made the following revisions.
A few comments and questions are added:
1) It is usual that the Results and Discussion section is after the Materials and Methods section. I believe this is just a typographical error.
Answer: The basic format of Molecules is “the Materials and Methods section is after the Results and Discussion section”. We are following it.
2) The authors commented on the change in fusion enthalpy for the HDPE sample of less than 10%, except for the sample with 24 days of degradation before detachment. Do they have any explanation for this exception (value 120)? Why can we observe a decrease in crystallinity in this sample?
Answer: We added the following sentences in lines 166–170 on the 5 page: The reason for the significant decrease in melting enthalpy of the degraded sample on the 24th is unclear, but it is likely that there are parts that have degraded so much that even the crystalline parts that have a high melting point have been destroyed. If the degradation time is even longer, it is thought that these parts will no longer have a melting point and will not be observed.
3) The authors reported 2 different methods of synthesis (polymerization) of PE. Can the authors comment on whether there is any possible effect of the synthetic route used on the degradation process, other than the different final PE structures?
Answer: We believe that differences in the synthetic routes may have some impact on the degradation process. We do not know to what extent, or whether there is a clear enough difference to make a comparison, since various factors, such as the type of catalyst, storage conditions, etc., can affect the process.
4) For a better comparison, in addition to the two SEM photographs already shown in Figure 6, I would prefer to see an SEM photograph of the surface pits of the scale of 1 μm.
Answer: The SEM photograph in Figure 6 was taken more than three months ago, so in order to take a photograph of a hole on the scale of 1 μm, we would have to make a new sample and take all the photographs again. Furthermore, it is not always possible to take a clear photograph with the same scale using the SEM after it has been re-observed. What we want to emphasize in Figure 6 is that there is a part of the shape that corresponds to the shape of the pit (see line 197 on page 6). Therefore, we took SEM photographs of sizes that clearly show the shape. We don't think it's necessary to have the sizes all the same. We believe that these SEM photos provide sufficient evidence.
5) In line 208, the sentence “After 30 days of degradation, the zeta potential value was approximately −10 mv, which was 10 mv higher than the −20 mv value after 21 days” is repeated.
Answer: The corresponding sentence has been deleted. It's clearly a careless mistake. Thank you for pointing it out.
6)
7) The authors only showed the results of the degradation time dependence of Zeta potential and hydrodynamic size values (with or without surfactant) of the detachment LDPE sample after degradation. It would be nice to compare these dependencies with HDPE.
Answer: Thank you for your suggestions. It is certainly necessary to measure the zeta potential and investigate the change in hydrodynamic size with and without surfactant for HDPE samples in the same way as for LDPE samples. However, as shown in the paper, the preparation of HDPE nanoparticles alone has not been successful. The method used in this paper to produce the nanoparticles results in the inclusion of large microsized crushed particles due to the spherulite structure of HDPE. For this reason, even if you measure the zeta potential, etc., it is not possible to compare it with LDPE. At present, we are developing a new process that adds a mechanical processing method to produce nano-HDPE particles with relatively uniform particle distribution. In our next paper we hope to show you the HDPE data.
8) The authors mentioned that the pH values of the solution were changed from 8.2 to 3 during each set, can they better explain the reason and possibly use a citation for this procedure?
Answer: We added the following sentence in line 265–267 on page 8: The pH of the seawater was initially 8.2, and the SO4•− was gradually converted to SO42−, reducing the pH of the seawater solution to 3 at the time of the daily exchange [16].
9) The authors should explain the EDM abbreviation.
Answer: It is not “EDM”, but the “accelerated degradation method”. This has been corrected.
10) The authors should add the FTIR spectra to the SI section.
Answer: We have added FT-IR spectra of degraded HDPE and LDPE in Figures S2 and S3 to the SI section.
11) In line 275 it is written PP, there should probably be PE.
Answer: We rewrote it.
12) If the authors want to use HDPE and LDPE films to create NP models, what do they think about the fillers that are usually used in PE materials? Can they affect PE degradation or NP formation?
Answer: It is true that many PE materials actually used contain fillers. Depending on the type of filler, it can have a significant impact on the degradation of PE, and as a result, it can affect the rate of NP formation. In the next stage, we would like to investigate the impact of various fillers on the behaviour of NP formation.
13) The authors wrote about the addition of 1% Triton® X-114 to stabilize the NP particles. Can the authors comment on why they use a 1% solution? Do they have a reason for choosing this concentration? Did they choose this concentration based on some publication (please add the citation) or did they try multiple concentrations to see which concentration is already suitable for stabilization? Did the authors also try different surfactants? Did the authors filter the surfactant solution before the DLS measurement? Isn't the 1% solution too viscous?
Answer: We are also conducting research on degraded PS nanoparticles, and we have successfully dispersed them using 1% Triton® X-114 as a surfactant. Therefore, we also dispersed LDPE nanoparticles at the same concentration of 1%. We calculated the viscosity of the 1% solution and calculated the hydrodynamic size. The surfactant was completely dissolved in water, and no filtration was performed. Since the dispersion of LDPE nanoparticles was successful, we did not try to disperse them at other concentrations. Since Triton® X-114 is known to be ecotoxic, it is necessary to switch to a different surfactant, but at present we have not yet switched to a different surfactant. In the next paper, we plan to look for a surfactant.
Reviewer 2 Report
Comments and Suggestions for Authors
This is a great study evaluating the differences in nanoplastics formation behavior between HDPE and LDPE. The result showed the key parameters changes during degradation process. However, there are lacking quantitative analysis in concerning the amount or percent NPs formation. Is there a way to show this difference? There were some general comments showed below.
1. In the abstract, the full name of HDPE and LDPE need stated first.
2. In the result and discussion section, the figure information needs to be cited in the content part, to make a link between then word and the figure.
3. In figure 1, how many replicates used for analysis, is there statistic significance between treatments and time points.
4. The figure labels on figure 2 was from (C) and (D), which may need change to A, B. Besides, the mainly changes of the SEM may be labeled on the images, can help the reader to follow. The figure S1 can be merged into figure 1, with panels of A, B ,C and D. This will help to understand the changes under time factor. And it is not very big of Fig. 1. The scale bar on the images were not directly. I think using the same length with different numbers can help. Or write out the magnification times on each image. Why not use the same sized images?
5. What is the format of the figure 4? Is it mean+/-range? This information need add in the figure legend.
6. Figure 5 also need replicates and significance analysis.
Author Response
Reviewer 2
This is a great study evaluating the differences in nanoplastics formation behavior between HDPE and LDPE. The result showed the key parameters changes during degradation process. However, there are lacking quantitative analysis in concerning the amount or percent NPs formation. Is there a way to show this difference? There were some general comments showed below.
Answer: We appreciate the reviewer's appropriate comments. We have made the following revisions.
- In the abstract, the full name of HDPE and LDPE need stated first.
Answer: We revised the names (see L 63–64 on 3 page).
- In the result and discussion section, the figure information needs to be cited in the content part, to make a link between then word and the figure.
Answer: We understand it. We have linked the text and diagrams as much as possible. However, we are not used to creating layouts ourselves. For this reason, in some diagrams, the corresponding text is separated. We cannot adjust it any further.
- In figure 1, how many replicates used for analysis, is there statistic significance between treatments and time points.
Answer: The number of replicates used for the analysis was three samples each. The processing was also carried out simultaneously, and the measurements were also taken almost simultaneously (within about 30 minutes). With regard to statistical significance, this is a commonly used method for measuring the carbonyl index (CI) value of polymer, and is sufficient.
- The figure labels on figure 2 was from (C) and (D), which may need change to A, B. Besides, the mainly changes of the SEM may be labeled on the images, can help the reader to follow. The figure S1 can be merged into figure 1, with panels of A, B ,C and D. This will help to understand the changes under time factor. And it is not very big of Fig. 1. The scale bar on the images were not directly. I think using the same length with different numbers can help. Or write out the magnification times on each image. Why not use the same sized images?
Answer: We have rewritten (C) and (D) in Figure 2 as (A) and (B). The magnification has also been added to the captions of Figures 2 and S1 respectively. It is difficult to match the magnification because the size of the object we want to see is different!
We also think that the proposal to include Figure S1 of the supplement in Figure 1 of the main text would certainly make it easier to understand. However, if we include any more high-resolution SEM photographs, the file size will become too large to send. For this reason, we have separated it into Figure S1 in the supplement, taking into account its importance. It is difficult to replace the figure due to file size limitations.
- What is the format of the figure 4? Is it mean+/-range? This information need add in the figure legend.
Answer: Error bars indicate the difference from the mean. We added the sentence in the figure 4 caption.
- Figure 5 also need replicates and significance analysis.
Answer: Error bars indicate the difference from the mean. We added the sentence in the figure 5 caption.
Round 2
Reviewer 1 Report
Comments and Suggestions for Authors
Based on the answers and clarification of some points from the authors, I recommend the article for publication.